# Dynamic genome evolution and complex virocell metabolism of globally-distributed giant viruses

Mohammad Moniruzzaman[1], Carolina A. Martinez-Gutierrez[1], Alaina R. Weinheimer [1] &
Frank O. Aylward [1✉]

The discovery of eukaryotic giant viruses has transformed our understanding of the limits of viral complexity, but the extent of their encoded metabolic diversity remains unclear. Here we generate 501 metagenome-assembled genomes of Nucleo-Cytoplasmic Large DNA Viruses (NCLDV) from environments around the globe, and analyze their encoded functional capacity. We report a remarkable diversity of metabolic genes in widespread giant viruses, including many involved in nutrient uptake, light harvesting, and nitrogen metabolism. Surprisingly, numerous NCLDV encode the components of glycolysis and the TCA cycle, suggesting that they can re-program fundamental aspects of their host's central carbon metabolism. Our phylogenetic analysis of NCLDV metabolic genes and their cellular homologs reveals distinct clustering of viral sequences into divergent clades, indicating that these genes are virus-specific and were acquired in the distant past. Overall our findings reveal that giant viruses encode complex metabolic capabilities with evolutionary histories largely independent of cellular life, strongly implicating them as important drivers of global biogeochemical cycles.

---

[1] Department of Biological Sciences, Virginia Tech, Blacksburg, VA 24061, USA. ✉email: faylward@vt.edu

Nucleocytoplasmic large DNA viruses (NCLDV) are a diverse group of eukaryotic viruses that include several families known for both their large virion size, reaching up to 1.5 μm, and genomes reaching ~2.5 million base-pairs in length[1]. The discovery of the first giant virus, *Acanthaomeba polyphaga mimivirus*, led to a paradigm shift in the field of virology by showing that, contrary to the traditional view of viruses as filterable infectious agents, viruses could be larger than some cellular lineages both in terms of physical size and genomic contents[2,3]. Several subsequent studies have continued to expand our knowledge of NCLDV diversity through the discovery of *Pithoviridae*[4], *Marseilleviridae*[5], *Pandoraviruses*[6], and several other members of the *Mimiviridae* family[7–9], all encoding large genomes with diverse genomic repertoires[10], and evolutionary genomic analysis has revealed common ancestry of all these groups together with algal viruses of the family *Phycodnaviridae* and vertebrate viruses of the families *Iridoviridae*, *Poxviridae*, and *Asfarviridae*[11–13].

The large genomic repertoires of NCLDVs have sparked interest regarding the evolutionary processes shaping their genome evolution, diversity, and potential role in modulating ecological dynamics, which remain poorly understood. Many pioneering discoveries of NCLDV over the last decade have leveraged Amoebozoa as a model host for isolation[1], but it is likely that a variety of other unicellular eukaryotes in the environment are infected by these viruses. Some *Phycodnaviridae* members that infect algae have been studied for decades[14], and several algae-infecting Mimiviruses have recently been isolated from diverse aquatic systems[15,16]. Moreover, recent cultivation-independent analyses have provided tantalizing evidence suggesting that some NCLDV groups are broadly distributed in nature and are potentially playing critical roles in the ecological and evolutionary dynamics of unicellular eukaryotes, particularly in aquatic environments[17,18].

Understanding the phylogenetic and genomic diversity of NCLDV in the environment is especially critical given recent findings on the biogeochemical significance of virus-mediated metabolic reprogramming of host cells into virocells[19–21]. A virocell is defined as a cell undergoing lytic virus infection that has altered nutrient demands compared to a healthy cell and a distinct physiological trajectory geared towards virus production rather than cellular growth and propagation[22]. The large and dynamic genomes of NCLDVs have been shown to encode a variety of metabolic genes, including those involved in nitrogen metabolism[23], fermentation[16,23], and sphingolipid biosynthesis[19], which likely contribute to shifts in host physiology during infection. Although these genes are thought to be acquired by NCLDV from diverse sources through Lateral Gene Transfer (LGT), the origin of many of these genes and the extent to which they are characteristic of NCLDV genomes more broadly remains obscure. Given the emerging scientific consensus on their ability to rewire the physiology of globally-abundant protists and impact marine biogeochemistry[24], it is imperative to obtain a comprehensive view of the genome diversity, evolutionary dynamics, and potential metabolic activities of these 'giants' of the virosphere.

In this study we assess the genomic diversity and encoded metabolic diversity of NCLDV in the environment through large-scale generation of metagenome-assembled genomes and analysis of their functional capacity. We identify diverse metabolic genes in widespread giant viruses, including many involved in nutrient uptake and processing, light harvesting, and central nitrogen metabolism, underscoring the complex interplay between these viruses and their hosts. In addition, we report numerous giant viruses that encode components of glycolysis, gluconeogenesis, the glyoxylate shunt, and the TCA cycle, including one genome with a 70%-complete glycolytic pathway, suggesting that they can re-program fundamental aspects of their host's central carbon metabolism. Moreover, we present phylogenetic analysis demonstrating that viral metabolic genes have disparate evolutionary histories from their cellular homologs and were potentially acquired early in the evolution of NCLDV. Our findings reveal that globally-abundant giant viruses encode diverse metabolic capabilities with evolutionary histories distinct from cellular life, highlighting their genomic complexity and important roles in ecosystems around the globe.

## Results

**Phylogenetic diversity of NCLDV MAGs.** To address critical questions regarding the genomic diversity, evolutionary relationships, and virocell metabolism of NCLDVs in the environment, we developed a workflow to generate metagenome-assembled genomes (MAGs) of NCLDVs from publicly-available metagenomic data (see Methods, Supplementary Fig. 1). We surveyed 1545 metagenomes and generated 501 novel NCLDV MAGs from individual samples that ranged in size from 100 to 1400 Kbp. Our workflow included steps to remove contigs potentially originating from cellular organisms or bacteriophage, as well as to minimize possible strain heterogeneity in each MAG. To ensure our NCLDV MAGs represented nearly-complete genomes, we only retained MAGs that contained at least 4 of 5 key NCLDV marker genes that are known to be highly conserved in these viruses[11] and had a total length >100 Kbp (see Methods for details and rationale). Most of the MAGs were generated from marine and freshwater environments (444 and 36, respectively), but we also found 21 in metagenomes from bioreactors, wastewater treatment plants, oil fields, and soil samples (labeled "other" in Fig. 1, Supplementary Fig. 2; details in Supplementary Dataset 1).

We constructed a multi-locus phylogenetic tree of the NCLDV MAGs together with 121 reference genomes using five highly conserved genes that have been used previously for phylogenetic analysis of these viruses[11] (Fig. 1). The majority of our MAGs placed within the *Mimiviridae* and *Phycodnaviridae* families (350 and 126, respectively), but we also identified new genomes in the *Iridoviridae* (16), *Asfarviridae* (7), *Marseillviridae* (1), and *Pithoviridae* (1). The identification of a large number of *Mimiviridae* members in our study is consistent with previous analyses suggesting high diversity of this family in marine systems[18,25,26]. Our phylogeny revealed that the *Phycodnaviridae* are polyphyletic and consist of at least two distinct monophyletic groups, one of which is sister to the *Mimiviridae* (termed Late *Phycodnaviridae*, 108 MAGs), and one which is basal branching to the *Mimiviridae*-Late *Phycodnaviridae* clade (termed Early *Phycodnaviridae*, 18 MAGs). The monophyly of the combined *Phycodnaviridae* and *Mimiviridae* families has been suggested previously based on concatenated marker gene phylogenies[11,15,27], although one recent study reported an alternative topology in which the *Asfarviridae* also placed within this broader group[28]. In addition to the phylogeny, we evaluated the pairwise Average Amino Acid Identity (AAI) between NCLDV genomes to assess genomic divergence. AAI values provided results that were largely consistent with our phylogenetic analysis and showed that intra-family AAI values ranged from 26 to 100% (Fig. 1b), highlighting the substantial sequence divergence between even NCLDV genomes within the same family.

Given the large diversity within each of the NCLDV families, we sought to identify major clades within these groups that could be used for finer-grained classification. Using the rooted NCLDV phylogeny we calculated optimal clades within each family using the Dunn index[29] (see Methods), resulting in 54 total clades, including 18 from the *Mimiviridae*, 13 for the Early

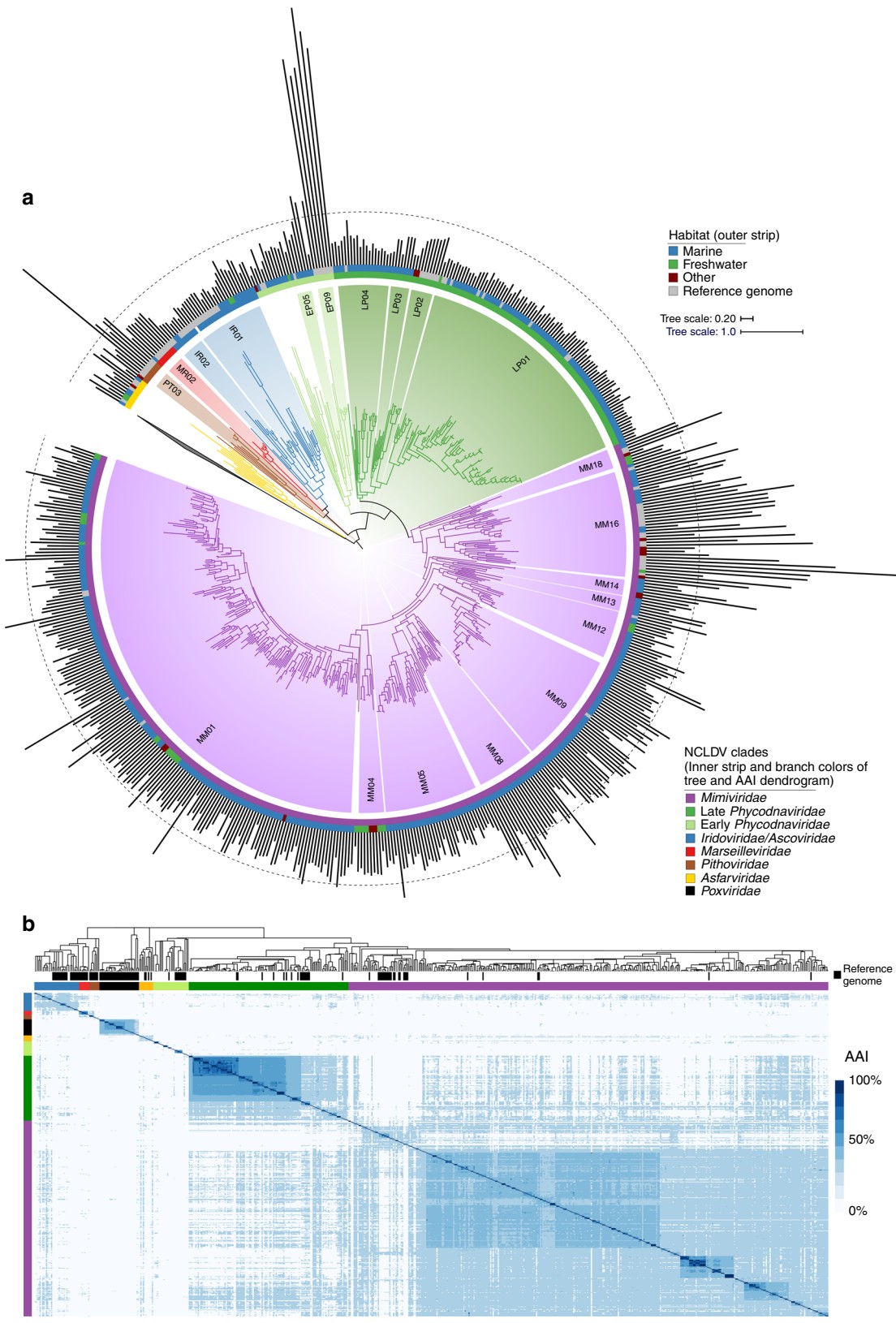

*Phycodnaviridae*, and 6 for the Late *Phycodnaviridae* (Fig. 1, Supplementary Dataset 1). No cultured representatives were present in 31 of the clades (57%), including 2 from the *Asfarviridae*, 9 from the Early *Phycodnaviridae*, 1 in the *Iridoviridae*, 3 in the Late *Phycodnaviridae*, 1 in the *Marseillevir-idae*, 14 in the *Mimiviridae*, and 1 in the *Pithoviridae*. Compared

to references available in GenBank, this greatly increases the number of available genomes in the *Mimiviridae* and *Phycodna-viridae*, highlighting the vast diversity of environmental NCLDV that have not been sampled using culture-based methods.

Analysis of the genome size distribution across the NCLDV phylogeny provided results that are consistent with the current

**Fig. 1 Evolutionary relationships of NCLDV MAGs and reference genomes. a** Phylogeny of the 501 NCLDV MAGs presented in this study together with 121 reference genomes. The phylogeny was constructed from a concatenated alignment of 5 highly conserved marker genes that are present throughout the NCLDV families using the VT + F + I + G4 model in IQ-TREE. The tree is rooted at *Poxviridae/Asfarviridae* branch, consistent with previous studies[11]. The inner strip is colored according to the phylogeny of the MAGs, while the outer strip is colored according to the habitat in which they were found. The bar chart represents genome size, which ranges from 100–2474 Kbp, and the dotted line denotes the 500 Kbp mark. Clades with >5 genomes are indicated with two letter abbreviations and clade numbers. MM: *Mimiviridae*, EP: Early *Phycodnaviridae*, LP: Late *Phycodnaviridae*, IR: *Iridoviridae*, MR: *Marseilleviridae*, PT: *Pithoviridae*. For the list of all the clades, see Dataset 1. **b** Average amino acid identity (AAI) heatmap of the MAGs and reference genomes, with rows and columns clustered according to the phylogeny.

knowledge of these viruses. For example, Late *Phycodnaviridae* clade 1 contained sequenced representatives of the Prasinoviruses, including known *Ostreococcus* and *Micromonas* viruses, which encode the smallest *Phycodnaviridae* genomes known[30]. Consistent with this finding, the MAGs belonging to this clade were also smaller and ranged in size from 100–225 Kbp, suggesting that small genome size is broadly characteristic of this group. By comparison, genomes in the Early *Phycodnaviridae* were larger and formed more divergent groups with long branches, suggesting a large amount of untapped diversity in this clade (Fig. 1a). The Pandoraviruses and *Mollivirus sibericum*, notable for their particularly large genomes, formed a distinct clade in the early *Phycodnaviridae*. The Coccolithoviruses and Phaeoviruses[31] were also placed in the Early *Phycodnaviridae*, and we identified 7 and 2 new members of these groups, respectively. Compared to the Late *Phycodnaviridae*, genome sizes of our MAGs were also notably higher in the *Mimiviridae*, which are known to encode among the largest viral genomes. In Mimivirus Clade 16, which includes *Acanthaeomeba polyphaga mimivirus*, we identified 19 new MAGs, 13 of which have genomes >500 Kbp. Taken together, these results are consistent with the larger genomes that have been observed in the *Mimiviridae* compared to the Late *Phycodnaviridae*[11]. Although our NCLDV MAGs contain most marker genes we would expect to find in these genomes, it is likely that many are not complete, and these genome size estimates are therefore best interpreted as underestimates.

**Evolutionary genomics of the NCLDV.** To assess the diversity of protein families across the NCLDV families, we calculated orthologous groups (OGs) between our MAGs and 126 reference genomes, resulting in 81,411 OGs (Supplementary Dataset 2). Of these, only 21,927 (27%) had detectable homology to known protein families in the EggNOG, Pfam, TigrFam, and VOG databases, highlighting the large number of novel genes in NCLDV genomes that has been observed in other studies[32,33]. Moreover, 55,692 (68%) of the OGs were present in only one NCLDV genome (singleton OGs), and overall the degree distribution of protein family membership revealed only a small number of widely-shared protein families (Fig. 2a, b), consistent with what has been shown for dsDNA viruses in general[34]. To visualize patterns of gene sharing across the NCLDV we constructed a bipartite network in which both genomes and OGs can be represented (Fig. 2c). Analysis of this network revealed primarily family-level clustering, with the *Mimiviridae* and early and late *Phycodnaviridae* clustering near each other, and the *Pithoviridae*, *Marseiviridae*, and *Poxviridae* clustering separately. Interestingly, although Pandoraviruses are members of the Early *Phycodnaviridae* clade, they clustered independently in a small sub-network, indicating that the particularly large genomes and novel genomic repertoires in this group are distinct from all other NCLDVs. These patterns suggest that genomic content in the NCLDV is shaped in part by evolutionary history, but that large-scale gains or losses of genomic content can occur over short evolutionary timescales, similar to what has occurred in the Pandoraviruses. This indicates that over long evolutionary

timescales the genome evolution of NCLDV is shaped by a mixture of vertical inheritance and LGT, in many ways at least qualitatively similar to that of *Bacteria* and *Archaea*[35].

To further elucidate the evolutionary history of the large number of genes in NCLDVs, we investigated clade-specific patterns in gene sharing. We found distinct clustering of NCLDV OGs based on their presence in NCLDV clades, indicating that the majority of the OGs are unevenly distributed across clades (Fig. 3a). This was confirmed by an enrichment analysis, where we identified sets of enriched OGs in each of the major NCLDV clades (Mann-Whitney U test, corrected *p*-value <0.01). The most common functional categories among the clade-specific OGs are predicted to be involved in DNA replication, translation, and transcription. Translational machinery was particularly enriched in Mimivirus clade 16, which contains many cultivated representatives known to have the highest proportion of translation-associated genes of any virus[36,37]. The clade-specific genomic repertoires of NCLDV suggest that this is an appropriate phylogenetic scale for examining functional diversity across the NCLDV, and we anticipate these clades will be useful groupings that can be used in future studies examining spatiotemporal trends in viral diversity in the environment.

**Metabolic potential of the NCLDV.** Relatively recent studies on model NCLDV-host systems have pointed out the presence of genes involved in rewiring key aspects of cell physiology during infection, such as apoptosis, nutrient processing and acquisition, and oxidative stress regulation[23,38–40]. We found a number of genes involved in such processes to be broadly encoded across NCLDVs, particularly in the *Mimiviridae* and *Phycodnaviridae* families (Fig. 3b). Superoxide dismutase (SOD) and Glutathione peroxidase (GPx), key players in regulating cellular oxidative stress, are prevalent in phylogenetically divergent NCLDVs. Giant virus replication potentially occurs under high oxidative stress inside the host cells[40] and thus, the presence of enzymes with antioxidant activity might be crucial in preventing damage to the viral machineries. SOD was biochemically characterized in *Megavirus chilensis*, and was suggested to reduce the oxidative stress induced early in the infection[39]. In addition, GPx was found to be upregulated during infection by algal giant viruses[39,40]. Genes putatively involved in the regulation of cellular apoptosis are also widespread in giant viruses, including C14-family caspase-like proteins and several classes of apoptosis inhibitors, such as Bax1[41]. C14-family metacaspases were reported in a giant virus obtained through single virus genomics approach, while viral activation and recruitment of cellular metacaspase was found during *Emiliania huxleyi* virus (*EhV*) replication[38,39]. In *Chlorella* viruses, a K+ channel (KcV) protein mediates host cell membrane depolarization, facilitating genome delivery within the host[42]. We identified KcV in genomes from all the major clades of late *Phycodnaviridae* and *Mimiviridae*, suggesting that host membrane depolarization is a widely-adopted aspect of NCLDV infection strategy. Lastly, in almost all the major *Mimiviridae* and *Phycodnaviridae* clades we detected genes involved in DNA repair and processing, such as photolyases,

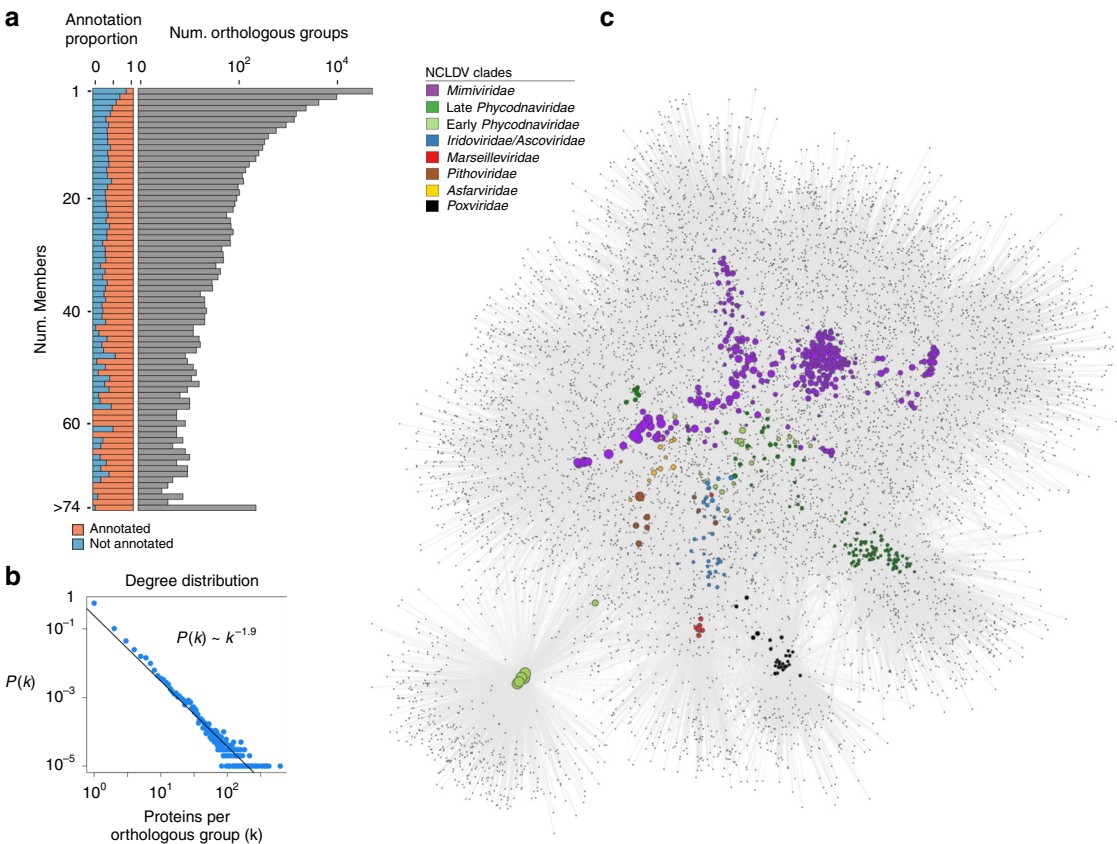

**Fig. 2 Gene-sharing patterns of the NCLDV MAGs and reference genomes. a** The distribution of the orthologous groups (OGs) in the NCLDV MAGs and reference genomes. The barplot on the left shows the proportion of OGs in each frequency category that could be assigned an annotation, while the barplot on the right shows the total number of OGs in each frequency category (log scale). **b** The degree distribution of the OG occurrence in the genomes analyzed. The best fit to a power law distribution is also shown. **c** A bipartite network of the OGs, with large nodes corresponding to genomes and small nodes corresponding to OGs. The size of the genome nodes is proportional to their genome size, and they are colored according to their family-level classification.

mismatch repair (*mutS*), histones, and histone acetyl transferases, of which the latter two have previously been reported in a number of giant virus families, with a possible role of viral histones in packaging of DNA within the capsid[9,15,43,44]. All together, these results demonstrate that many important aspects of viral reproduction and infection found in cultivated NCLDV are widespread in nature and a common feature of virocell metabolism during giant virus infection.

Viruses are thought to restructure host metabolism during infection to align with virion production rather than cell growth, leading to altered nutrient demands inside the cell[20,45]. We found that genes involved in nutrient acquisition and light-driven energy generation are widespread in several NCLDV clades, including rhodopsins, chlorophyll a/b binding proteins, ferritin, central nitrogen metabolism, and diverse nutrient transporters (Fig. 3b), consistent with other studies that have observed some of these genes in diverse NCLDV genomes[23,46,47]. In addition, studies on the structure and mechanism of rhodopsin present in two giant viruses have revealed that these are light-driven proton pumps, with potential to reshape energy transfer within the infected host[48,49]. Similarly, widely-distributed chlorophyll a/b binding proteins in giant viruses might increase photosynthetic light-harvesting capacity of infected cells, since protists and plants are known to suppress their photosynthetic machineries, including the chlorophyll binding antenna proteins, in response to virus infection[39,50,51]. In addition, the presence of the key eukaryotic iron storage protein ferritin[52] and transporters predicted to target ammonium (AmT), phosphorus (Phosphate permease and

Phosphate:Na+ symporters), sulfur (TauE/SafE family), and iron (Fe2+/Mn2+ transporters) highlights the shifting nutrient demands of virocells compared to their uninfected counterparts. Most of the MAGs were found in aquatic environments where nutrient availability may be limiting for cellular growth, and alteration of nutrient acquisition strategies during infection may be a key mechanism for increasing viral production. For example, although iron is crucial for photosynthesis and myriad other cellular processes[53], it is often present in low concentrations in marine environments[54,55], and the production of viral ferritin may aid in regulating the availability of this key micronutrient during virion production. Moreover, nitrogen and phosphorus are limiting for microbial growth in many marine ecosystems, and given the N:C and N:P ratios of viral biomass are relatively higher than that of cellular material[56], it is likely crucial for viruses to boost acquisition of these nutrients with their own transporters. Indeed, a recent study has revealed that an NCLDV-encoded ammonium transporter (AmT) can influence the nutrient flux in host cells by altering the dynamics of ammonium uptake[23].

Strikingly, many NCLDV genomes encode genes involved in central carbon metabolism, including most of the enzymes for glycolysis, gluconeogenesis, the TCA cycle, and the glyoxylate shunt (Figs. 3b, 4a, Supplementary Fig. 5). Central carbon metabolism is generally regarded as a fundamental feature of cellular life, and so it is remarkable to consider that giant viruses cumulatively encode nearly every step of these pathways. These genes were particularly enriched in Mimivirus clades 1, 9, and 16,

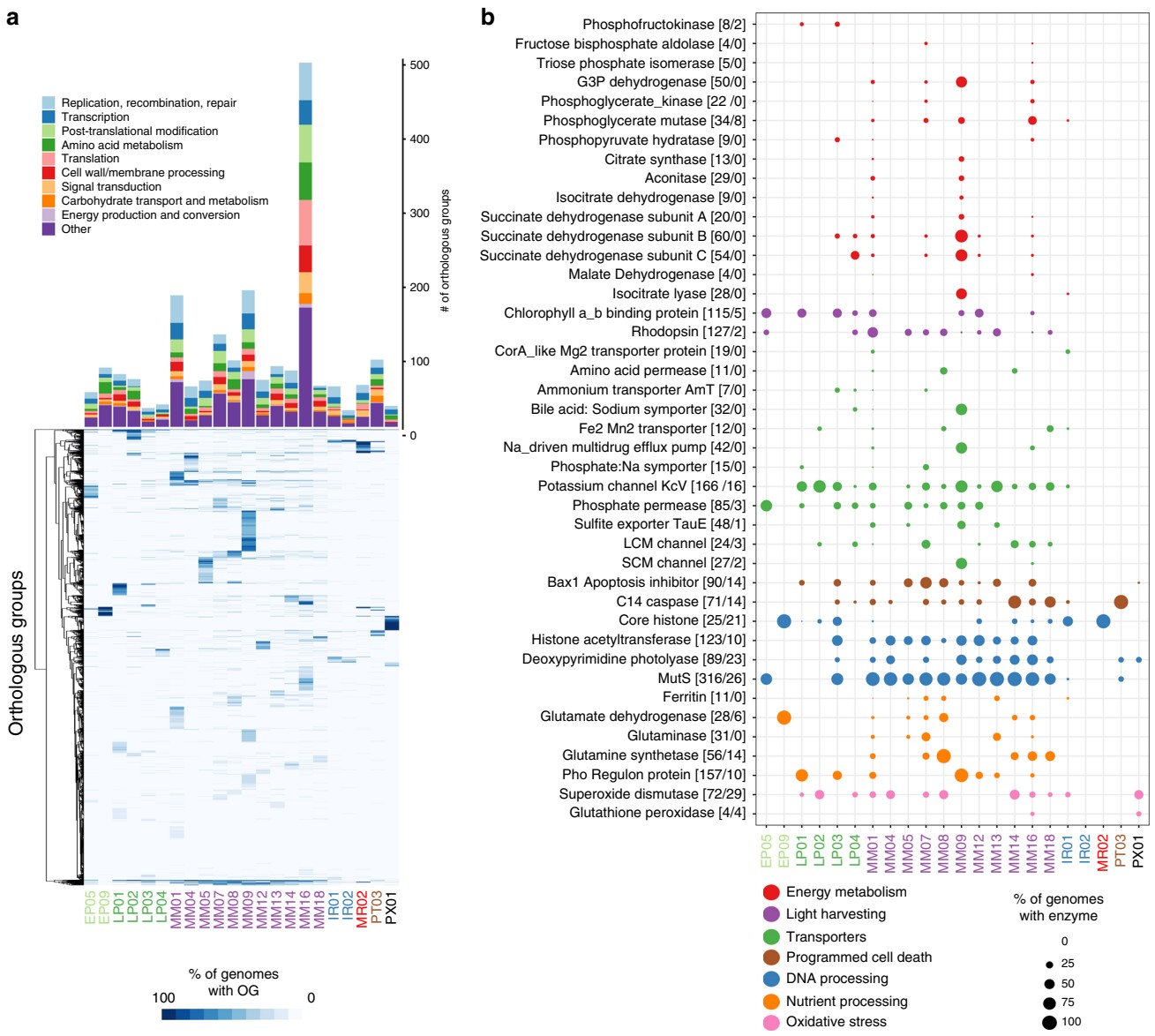

**Fig. 3 Distribution of orthologous groups and metabolic enzymes in NCLDV clades. a** The barplot shows the number of enriched OGs in each of the major NCLDV clades analyzed in this study. Only a subset of total functional categories are shown here; a full table can be found in Supplementary Dataset 2. The heatmap shows the occurrence of OGs with >5 total members across the major NCLDV clades, with shading corresponding to the percent of MAGs in that clade that encode a given OG. **b** A bubble plot of select metabolic genes detected in the NCLDV clades, with bubble size proportional to the percent of genomes in a clade that encode that protein. The numbers in brackets next to each enzyme name denote the number of these proteins observed in the MAGs we present here and the number observed in reference NCLDV genomes, respectively. G3P: glycerol-3-phosphate; LCM: Large conductance mechanosensitive; SCM; small conductance mechanosensitive.

but a few of them were also present in several *Phycodnaviridae* members (Figs. 3b, 4a). The glycolytic enzymes glyceraldehyde-3-phosphate dehydrogenase (G3P), phosphoglycerate mutase (PGM), and phosphoglycerate kinase (PGK), as well as the TCA cycle enzymes aconitase and succinate dehydrogenase (SDH) were particularly prevalent. In addition, we identified a fused gene in 16 MAGs that encodes the glycolytic enzymes G3P and PGK, which carry out adjacent steps in glycolysis (Fig. 4a, b), representing a unique domain architecture that has not been reported in cellular lineages before. Interestingly, in many MAGs, TCA cycle genes were co-localized on viral contigs, suggesting possible co-regulation of these genes during infection (Fig. 4c). Remarkably, one NCLDV MAG (ERX552257.96) encoded enzymes for 7 out of 10 steps of glycolysis (Fig. 4d), highlighting the high degree of metabolic independence that some giant

viruses can achieve from their hosts. The fact that viruses encode these components of diverse central metabolic pathways underscores their potential to fundamentally reprogram virocell metabolism through manipulation of intracellular carbon fluxes.

Although the prevalence of genes involved in central carbon metabolism in the NCLDV MAGs strongly implicates them in modulating host metabolism, it is unclear at this point if these enzymes function in the same physiological context as their corresponding host versions. For example, succinate dehydrogenase has an important role in modulating cellular oxidative damage[57], and could have a similar function during NCLDV propagation, which is carried out within a highly oxidative cellular environment. Moreover, in a recent study G3P was directly implicated in starvation-induced negative regulation of vesicle formation in the Golgi and several other cellular transport

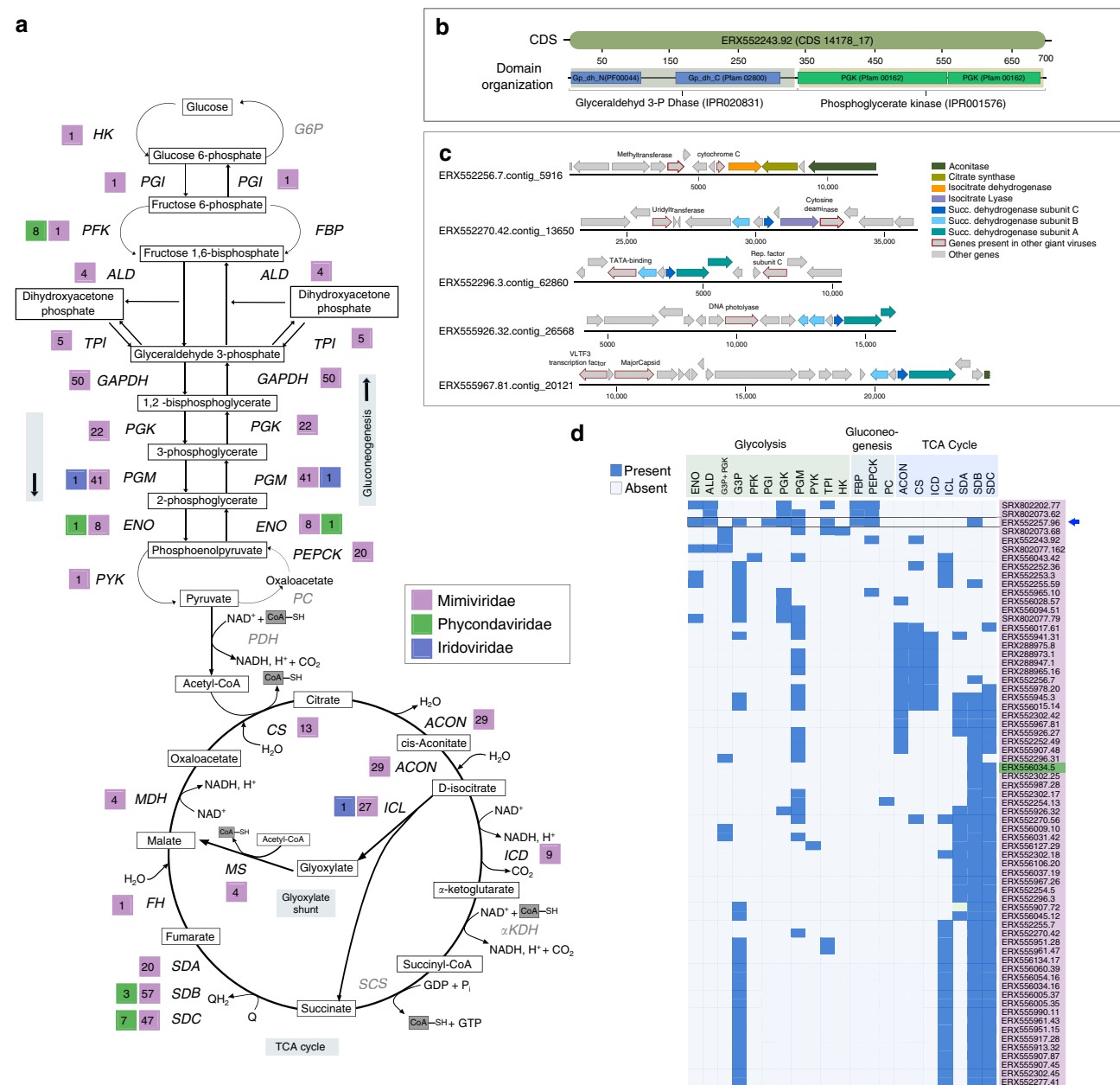

**Fig. 4 Presence of central carbon metabolic enzymes in the NCLDVs. a** Schematic of Glycolysis, Gluconeogenesis, and the TCA cycle, with the number of NCLDV MAGs harboring a particular enzyme provided beside abbreviated enzyme names. Enzymes that were not detected in any of the studied NCLDVs are in gray. **b** Representative CDS from genome ERX552243.92 illustrating the domain organization (PFAM and Interpro) of the fused-domain gene (G3P + PGK) involved in glycolysis, that was detected in 16 of the NCLDV MAGs. **c** Example of co-localization of genes involved in TCA cycle on genomic contigs from five representative NCLDV MAGs. Location of a number of other genes commonly present in NCLDVs are also shown. **d** Presence/absence of genes involved in central-carbon metabolism in NCLDV genomes assembled in this study. Only the genomes harboring 3 or more enzymes are shown. G3P + PGK indicates the fused-domain gene illustrated in panel B. Blue arrow indicates the genome that harbors 7 out of 10 enzymes involved in glycolysis. HK: hexokinase, PGI: Phosphoglucoisomerase, PFK: Phosphofructokinase, ALD: aldolase, TPI: Triose-phosphate isomerase, G3P: Glyceraldehyde 3-phosphate dehydrogenase, PGK: Phosphoglycerate kinase, PGM: Phosphoglycerate mutase, ENO: Enolase, PYK: Pyruvate kinase, PEPCK: PEP carboxykinase, FBP: Fructose 1,6-bisphosphatase, G6P: Glucose 6-phosphatase, PDH: Pyruvate dehydrogenase, PC: Pyruvate carboxylase, CS: Citrate synthase, ACON: Aconitase, ICL: Isocitrate lyase, ICD: Isocitrate dehydrogenase, αKDH: α-ketoglutarate dehydrogenase, SCS: Succinyl-CoA synthetase, SD: Succinate dehydrogenase (subunits A, B and C), FH: Fumarate hydratase, MS: Malate synthase, MDH: Malate dehydrogenase.

pathways independent of glycolysis[58]. It was proposed that by reducing energy consumption during starvation, G3P plays a complementary role in energy homeostasis alongside autophagy, which, in contrast, increases energy availability. Although the modulation of autophagy during NCLDV infection remains to be elucidated, it is possible that viral G3P could help avoid the lethal

consequences of starvation in the hosts, while autophagy-mediated recycling of proteins could make amino acids and other nutrients available. This possibility is further strengthened by the fact that a large number of NCLDV MAGs harbor phosphate starvation inducible protein (PhoH) and nutrient transporters which might work in concert with the G3P-mediated

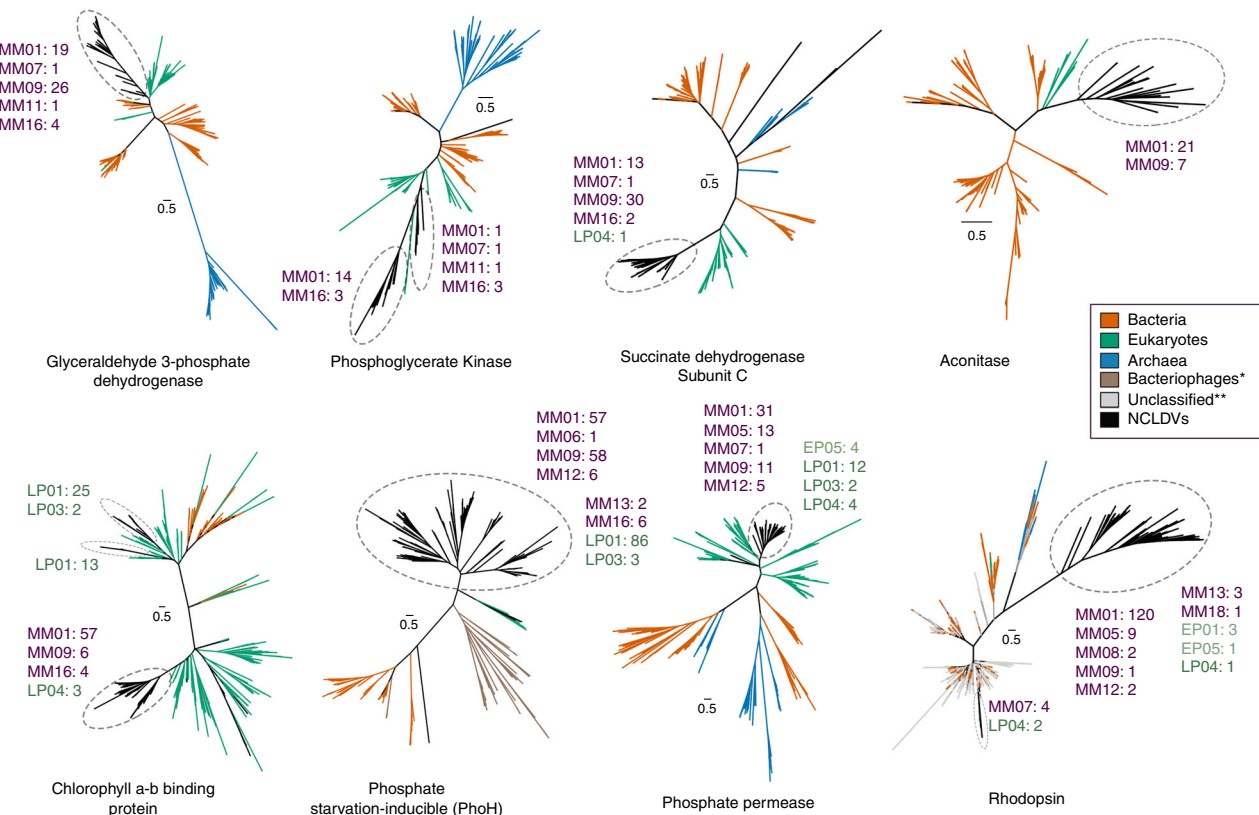

**Fig. 5 Phylogenetic reconstruction of a number of representative NCLDV genes likely involved in carbon and nutrient metabolism and light harvesting.** NCLDV-specific clusters are encircled with dashed ovals in each of the trees, while number of genes from different NCLDV-clades contributing to these monophyletic groups are also provided (MM: *Mimiviridae*, EP: Early *Phycodnaviridae*, LP: Late *Phycodnaviridae*) Colors of the clade names correspond to those in Fig. 1. Although node support values are not provided for better visual clarity, all the NCLDV-specific nodes are supported by >90% ultrafast bootstrap values (see Methods and Data availability statement for details). One asterisk denotes bacteriophage sequences, which are only present in the PhoH tree. Double asterisks denote unclassified sequences (environmental), which are only present in the rhodopsin tree.

mechanism to ensure virus propagation within the energy-limited host cells.

**Evolutionary history of NCLDV metabolic genes.** Phylogenies of a number of viral metabolic genes identified here together with their cellular homologs revealed that NCLDV sequences tended to group together in deep-branching clades, except for a few cases where multiple acquisitions from cellular sources was evident (Fig. 5, Supplementary Figs. 6–15). For example, aconitase, succinate dehydrogenase subunits B and C, PhoH, glyceraldehyde-3-phosphate dehydrogenase, and superoxide dismutase all showed distinct deep-branching viral clusters and were present in members of multiple NCLDV families, suggesting they diverged from their cellular homologs in the distant past (Fig. 5, Supplementary Figs. 6–15). This pattern was also observed for rhodopsin, similar to previous reports that NCLDV rhodopsins represent a virus-specific clade[48], although our study suggests that at least some NCLDVs independently acquired a bacterial rhodopsin. Phosphoglycerate kinase, chlorophyll a-b binding proteins, and ammonium transporter (AmT) also appear to have been acquired multiple times, but nonetheless show several deep-branching viral clades. These results demonstrate that while NCLDVs have acquired numerous central metabolic genes from cellular hosts, many of these metabolic genes have subsequently diversified into virus-specific lineages. Indeed, detailed functional characterization of viral rhodopsin and Cu-Zn superoxide dismutases has revealed that they have different structural and mechanistic properties compared to the cellular homologs[48,59], indicating that many metabolic genes in giant viruses evolved to have specific functions in the context of host-virus interactions. Our finding of the fused G3P-PGK glycolytic enzyme in many *Mimivirus* MAGs further reinforces this view and demonstrates that NCLDV are unique drivers of evolutionary innovation in metabolic genes.

These results run contrary to a canonical view of viral evolution in which viruses are seen as pickpockets that sporadically acquire genes from their cellular hosts rather than encoding their own virus-specific metabolic machinery[60]. Although these metabolic enzymes were likely acquired from cellular lineages at some point, their distinct evolutionary trajectory differentiates them from their cellular counterparts and demonstrates that NCLDV are themselves a driver of evolutionary innovation in core metabolic pathways. A recent study suggested that NCDLV have ancient origins and may even pre-date the last eukaryotic common ancestor[28], indicating there has been a long period of co-evolution between these viruses and their hosts during which these gene acquisitions could have taken place.

**Discussion**
Viruses have historically been viewed as accessories to cellular life, and as such their influence on biogeochemical cycles has largely been viewed through the lens of their impact on host mortality, rather than any direct metabolic activities of their own. The large number of cellular metabolic genes encoded in NCLDV genomes that we reveal in this study brings to light an alternative view in which virus-specific enzymes have a direct role in shaping virocell physiology. Scaled across viral infections in global aquatic environments, this raises the possibility that viral enzymes can

substantially alter global biogeochemical fluxes in their own right. Moreover, the distinct evolutionary lineages of viral metabolic genes implicate NCLDV as unique drivers of metabolic innovation, in stark contrast to the traditional view in which they are merely occasional pickpockets of cellular genes rather than de facto evolutionary innovators. Taken together, these findings argue that just as microbes are considered the engines that shape global biogeochemical cycles[61], viruses should be considered alongside their cellular counterparts as agents of metabolic fluxes with their own encoded physiology.

## Methods

**Assembling NCLDV genomes from metagenomes**. Although phylogenetic binning of metagenomic contigs belonging to Archaea and Bacteria is now commonplace[62], this approach is rarely used for viruses, and it was therefore necessary for us to develop a novel workflow to recover high-confidence NCLDV genomes from metagenomic data. Moreover, methods for assessing the completeness and potential contamination of prokaryotic bins, such as employed by the popular programs CheckM or Anvi'o[63,64], rely on knowledge of shared single-copy protein families in different lineages, but this information is not applicable to viral bins given their fundamentally distinct genomic repertoires. We therefore also developed a workflow for quality-checking NCLDV bins. The overall process can be divided into three main stages: (1) initial binning of contigs, (2) identification of bins corresponding to NCLDV, and (3) quality-checking bins to ensure contamination is not present. An overview of this workflow can be found in Supplementary Fig. 1.

**Initial binning of contigs**. We obtained assembled contigs (>10 Kbp) and coverage files for 1545 metagenomes that had been previously assembled in a large-scale study that examined bacterial and archaeal diversity[65]. We chose to use the program MetaBAT2[66] v. 2.12.1 for binning because this program bins contigs based on sequence coverage and tetranucleotide frequencies, which are metrics that would be expected to be consistent in viral genomes and therefore useful for binning. Moreover, although some binning tools rely on marker gene sets that are specific to cellular lineages, MetaBat2 does not use marker genes for binning and is therefore more appropriate for binning viral sequences. We used the parameters -s 100000, -m 10000, –minS 75, –maxEdges 75, which are more stringent than the default parameters and would be expected to yield more conservative, high-confidence binning results.

**Screening bins**. After binning contigs it is necessary to screen the bins to identify ones that correspond to putative NCLDV genomes. It has previously been shown that 5 highly conserved and NCLDV-specific protein families are present in almost all known NCLDV genomes, and we therefore used these for screening. The protein families correspond to the Major Capsid Protein (MCP), Superfamily II helicase (SFII), Virus-like transcription factor (VLTF3), DNA Polymerase B (PolB), and packaging ATPase (A32). We predicted protein sequences from all bins using Prodigal[67] v. 2.6.3 (default parameters), and matched proteins to the 5 NCLDV marker genes using HMMER[68] v. 3.2.1 with custom Hidden Markov Models (HMMS, see section "protein families used for screening" below). We only considered bins that had >= 4 of the markers for further analysis in order to exclude NCLDV genomes that were mostly incomplete. Moreover, we only considered bins >100 Kbp on the grounds that the smallest NCLDV genome is 103 Kbp[69], and that a higher cutoff may bias recovery against NCLDV clades with smaller genome sizes. After implementing these screens we recovered 517 candidate bins.

**Quality-checking bins**. To ensure that the bins were indeed viral and did not include contamination for cellular sources, we screened all contigs using a custom Python script called ViralRecall (https://github.com/faylward/viralrecall). This tool compares encoded proteins to virus-specific (from the VOG database) and cellular-specific HMMs to assess their provenance, and is therefore useful in determining contigs that may represent contamination from a cellular organism. ViralRecall uses custom subsets of the Viral Orthologous Groups (VOG: vogdb.org) and Pfam databases[70] for the virus-specific and cellular-specific HMMs, respectively, and generates a score for each contig (negative scores indicating more hits to cellular HMMs, positive scores indicating more hits to viral HMMs). In addition to ViralRecall we also used LAST[70,71] v. 959 (parameters -m 500) to compare all of the encoded proteins in each contig to RefSeq 92[72], and recorded the top 5 hits for each protein. To remove contigs that derived from cellular organisms, we removed contigs that had a ViralRecall score < 0 (indicting a net cellular signal), contained <3 encoded proteins with hits to HMMs in the VOG database, and had no LAST hits to known NCLDV proteins among the top 5 hits. In addition, to exclude possible bacteriophage sequences, we removed contigs for which the encoded proteins had at least one LAST hit to a bacteriophage and zero hits to a known NCLDV genome. A summary of all of the contigs removed in this way can be found in Supplementary Dataset 1.

To uncover strain heterogeneity, we identified cases where marker genes were found in multiple copies. We used SFII, VLTF3, A32, and PolB, for this, excluding MCP because multiple copies of this gene is commonplace in NCLDV genomes. We identified 16 bins where more than one marker gene was found to be present in multiple copies, and excluded these bins from further analysis. Of the remaining 501 bins, 62 contained one marker gene that was not single-copy, but these were retained because some complete NCLDV genomes contain multiple copies, and overly strict thresholds would exclude potentially novel NCLDV lineages with genomic repertoires distinct from what has been observed. Overall 501 bins passed all screening procedures and are subsequently referred to as NCLDV Metagenome Assembled Genomes (MAGs). Overall, by using strict binning parameters and excluding possible cellular or bacteriophage contigs, we recovered high quality MAGs that consisted of relatively few contigs (422 bins with < 20 contigs, including 2 bins with only a single contig; mean N50 contig size of 37.4 Kbp across MAGs).

**Protein families used for screening**. To screen preliminary bins and identify metagenome-assembled NCLDV genomes, we used a custom set of HMMs created for 5 NCLDV-specific protein families: The Major Capsid Protein (MCP), Superfamily II helicase (SFII), Virus-like transcription factor (VLTF3), DNA Polymerase B (PolB), and packaging ATPase (A32). These 5 protein families have previously been used for phylogenetic analysis of NCLDV and are typically not found in cellular organisms[11]. To generate these models, we manually annotated proteins from 126 complete NCLDV genomes available in NCBI that span the 7 major families. We then generated model-specific HMMER3 score cutoffs based on the scores recovered from matching known protein family members to these HMMs (Supplementary Fig. 4). These scores were used in determining the presence/absence of these protein families in the MetaBAT2 bins.

**Phylogenetic reconstruction of NCLDV MAGs**. To assess the phylogeny of the NCLDV MAGs we generated a concatenated tree of all 501 MAGs together with 121 reference NCLDV genomes using the marker genes PolB, VLTF3, MCP, A32, and SFII. These proteins have previously been shown to be useful for phylogenetic analysis of NCLDV. In some cases, NCLDV are known to have introns or split genes, and we generated a Python script to identify these cases, check to ensure the proteins hit to the same HMM and had no sequence overlap, and subsequently concatenate the proteins (code available at github.com/faylward/ncldv_markersearch). Alignments were created using ClustalOmega, and trimAl was used for trimming (parameter -gt 0.1). We ran IQ-TREE[73] v. 1.6.6 with the "-m TEST" ModelFinder option[74], which identified VT+F+I+G4 as the optimal model. We then ran IQ-TREE on the alignment with 1000 ultrafast bootstraps to assess confidence[75]. Although we used a set of 5 highly conserved markers that have been shown to reliably reconstruct NCLDV evolutionary relationships[11], other recent studies have used other markers such as DNA-dependant RNA Polymerase (RNAP) subunits and transcription elongation factor II-S (TFIIS)[28], and the observed evolutionary relationships between NCDLV groups may be affected by marker gene choice.

**Clade delineation**. Given the large phylogenetic diversity of NCLDV examined in this study, we sought to identify clades of closely related viruses within each of the major families. To this end we used the Dunn index to identify optimal clade-level delineations in our multi-locus phylogenetic tree of the NCLDV. We first generated a rooted ultrametric phylogenetic tree in R using the "ape" package and generated clades at different tree cut heights. For each cut height we calculated the Dunn index[29] using the "cluster.stats" package in R. We found that a height of 2.45 had the lowest Dunn index and therefore provided the most appropriate clustering (Supplementary Fig. 3). All clusters were then manually inspected and edited to ensure that they represented monophyletic groups, and these final clusters were used as clades. To confirm the clade-level distinctions were consistent across different evolutionary models, we also constructed a phylogenetic tree from the same concatenated protein alignment using the popular LG+I+G4 model and confirmed that our clades still formed monophyletic groups.

**Generation of orthologous groups and annotation**. We used ProteinOrtho v6.06[76] to calculate the orthologous groups shared between the 501 NCLDV MAGs and 127 reference genomes. Protein files were generated using Prodigal with default parameters. Because of the accelerated evolutionary rate of viruses, we used the relaxed parameters "–e = 1e−3 –identity = 15 –p = blastp+ –selfblast –cov = 40" for proteinortho. This resulted in 81,412 orthologous groups (OGs). For each OG, we randomly selected a representative for annotation. Representatives were compared to the EggNOG 4.5, TIGRFam v. 13.0, Pfam v. 31, and VOG (vogdb.org, accessed 10/01/2019) databases using HMMER3 (e-value 1e−5), and best his were recorded.

**Bipartite network analysis**. Bipartite networks of NCLDV genomes and their protein families were created in igraph[77]. For visualization purposes, only protein families present in >5 genomes were analyzed. In the bipartite graph two node types were present: Genome nodes and Protein Family nodes. Each protein family node was connected to a genome node if it was encoded in that genome. The

spring-directed layout was generated using the layout.fruchterman.reingold() command with 10,000 iterations.

**Average amino acid identity calculation**. AAI between the NCLDV genomes was calculated using a custom Python script available on GitHub (https://github.com/faylward/lastp_aai). The script employs pairwise LAST searches[71] (parameter -m 500; LAST version 959) of protein sequences and calculates the AAI and alignment fraction (AF) between all genome pairs. For visualization purposes, genome pairs in which one member had an AF < 10 were considered to have an AAI of 0.

**Clade-specific OG enrichment**. To evaluate if specific NCLDV clades were enriched in particular OGs, we performed an enrichment analysis on all clades with >5 members. For each OG, a Mann-Whitney U test was performed on overall OG membership in that clade compared to all other NCLDV (including reference genomes). Only OGs with a known annotation that were present in >6 genomes were used. P-values were corrected using the Benjamini-Hochberg procedure in R[78], and values <0.01 were considered significant.

**Phylogenetic reconstruction of NCLDV metabolic genes**. We collected the reference sequences for most of gene trees from the EggNOG database[79] with the following exceptions: for phosphate permease and chlorophyll a/b binding proteins we used sequences from the Pfam database[70], while rhodopsin sequences were collected from the MicRhoDE, a dedicated server for rhodopsins from different domains of life and environments[80]. In case of the superoxide dismutase (SOD) and phoH genes, we curated additional sequences from viruses other than NCLDVs from the NCBI Refseq database. For each gene, a diagnostic tree was built using FastTree[81] implemented in the ETE3 package. The diagnostic trees were used to select a smaller set of reference sequences and remove short or redundant sequences. For construction of the final trees, we aligned the sequences using ClustalOmega and trimmed using trimAl (parameter -gt 0.1). IQ-TREE[73] was used to build maximum likelihood phylogenetic trees with the model 'LG+I+G4' and 1000 ultrafast bootstrap replicates[75].

**Reporting summary**. Further information on research design is available in the Nature Research Reporting Summary linked to this article.

## Data availability
Supplementary Dataset 1, Supplementary Dataset 2, nucleotide sequences of NCLDV MAGs, predicted proteins, alignments used for phylogenies, raw orthologous groups membership files, and other major data products are available on the Aylward Lab Figshare account: https://figshare.com/projects/NCLDV/71138. The nucleotide and predicted protein sequences of the NCLDV MAGs can also be found on the Virginia Tech VTechData archives at: https://data.lib.vt.edu/files/m613mx716.

## Code availability
In this manuscript we used several custom Python scripts that can be accessed on GitHub. The ViralRecall script used for identifying viral regions in genomic data can be found at github.com/faylward/viralrecall. The NCLDV marker gene identification script can be found at github.com/faylward/ncldv_markersearch. The LAST-based average amino acid identity script can be found at https://github.com/faylward/lastp_aai.

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

## Acknowledgements

We acknowledge the use of the Virginia Tech Advanced Research Computing Center for bioinformatic analyses performed in this study. This work was supported by grants from the Institute for Critical Technology and Applied Science at Virginia Tech, an Alfred P. Sloan Research Fellowship, a Simons Early Career Investigator Award in Marine Microbial Ecology and Evolution (Grant No. 620443), and NSF IIBR-1918271 grant to F.O.A.

## Author contributions

F.O.A. and M.M. designed the experiments, F.O.A., M.M., C.A.M.G. and A.R.W. performed the experiments, F.O.A. and M.M. wrote the manuscript.

## Competing interests

The authors declare no competing interests.
