## [Peer Review File · Nature Communications]

Reviewers' comments:

Reviewer #1 (Remarks to the Author):

The authors describe important and novel work in the role of nucleocytoplasmic large viruses (NCLDV) in driving host metabolic processes across a wide range of pathways including nutrient uptake, light-harvesting, carbon metabolism, and nitrogen metabolism. The work is well-grounded in the viral literature and presents a new bioinformatics workflow for binning and distinguishing NCLDV contigs from the cellular metagenomes from which the contigs are derived. Moreover, careful attention is given to quality control, bioinformatics approaches and tool parameter choices for detecting viruses, making the findings compelling and grounded in a solid framework. The work greatly expands the number of available giant virus genomes, and also our understanding of the various clades of giant viruses and their evolutionary context. Finally, the results provide clear evidence for the role of giant viruses as drivers for biogeochemical cycles based on metabolic genes acquired from their hosts in the distant past. The manuscript is clearly written and offers new insight into the global impact of giant viruses that will clearly be well-cited and widely read by a broad audience. I have a few minor points below, and some points to consider in the discussion of the role of giant viruses in the carbon cycle based on my interpretation of the

data that the authors may want to consider.

Minor points:

Line 128: A reference is needed after “that has been observed in other studies”

Line 411: The authors need to have a reference to where the python script is located in Github to make the code available to the community.

Line 432: The authors need to state which tool was used for comparing representative sequences to reference databases. An e-value cut off is listed by no reference to the tool.

Discussion of central carbon metabolism lines 209-226:

The authors show in Figure 3, 4, and S5 that NCLDVs play an important role in carbon metabolism and encode many enzymes in glycolysis, gluconeogenesis, the TCA cycle, and the glyoxylate shunt. However, based on recent literature, the authors may be missing some important and new mechanisms that these viruses may be using that have not been seen in bacteriophage. In particular, the authors should consider these data in conjunction with other findings that “giant virus replication may occur under high oxidative stress” line 164, and “recent studies on model NCLDV-host systems have pointed out ... genes involved in rewiring key aspects of cell physiology during an

infection such as apoptosis, nutrient processing, and acquisition, and oxidative stress regulation” (line 159).

First, Figure 4A shows an abundance of succinate dehydrogenase, that has been shown to be involved in the formation and elimination of reactive oxygen species (see <https://doi.org/10.1016/j.bbabi.2016.03.012>). Therefore the role of SDH may not be in central carbon metabolism, but rather in dealing with oxidative stress.

Second, Figure 3b, shows that MM09 has the most carbon metabolic genes (in particular SDH and GAPDH) and also has PhoH related to phosphate acquisition which may be due to limited nutrient availability. Under starvation conditions, GAPDH has recently been shown to play an important role in energy homeostasis (Nature. 2018 Sep;561(7722):263-267. doi: 10.1038/s41586-018-0475-6. Epub 2018 Sep 12). Quoting from that paper, “Mechanistically, this involves the activation of AMPK to induce the redistribution of cytosolic GAPDH to multiple membrane compartments, which then targets ARF GAPs to exert inhibition on different transport pathways (Fig 4q). As the cell encounters starvation in many settings, considerable effort has been devoted to elucidating mechanisms of cell survival during this daunting situation. Autophagy has emerged as a major mechanism 1. In the overall energy equation, autophagy acts on the supply side to increase energy availability. In contrast, the transport inhibition by GAPDH that we have uncovered acts on the demand side to reduce energy consumption. Despite being fundamentally distinct, the two mechanisms nevertheless funnel into a common goal of promoting energy homeostasis during starvation. Another notable distinction between the two mechanisms is that the transport inhibition by GAPDH occurs more rapidly than autophagy. This more rapid mechanism likely allows the cell to stave off the lethal consequences of starvation while longer-term solutions, such as autophagy, are being mobilized.” Therefore, an alternative hypothesis is that NCLDVs produce GAPDH to “stave off the lethal consequences of starvation” in their hosts while autophagy mobilizes amino acids from the host that can be used for viral replication. This is further supported by the abundance of transport-related proteins encoded in these viruses, which may help to promote transport important for viral replication. This could be an important, and currently, an undescribed mechanism that the authors may want to consider especially given the genomic context that they have already laid out in the paper, and the fact that enzymes in C metabolism are not evenly distributed (Figure 4).

Reviewer: Bonnie Hurwitz

Reviewer #2 (Remarks to the Author):

In the manuscript entitled “Dynamic Genome Evolution and Complex Virocell Metabolism of Globally-Distributed Giant Viruses”, Moniruzzaman et al. generated 501 metagenome-assembled genomes of Nucleocytoplasmic large DNA viruses (NCLDV) from publically available metagenomes, using automatic binning taking advantage of sequence composition and within-sample coverage. Authors elected not to take advantage of differential coverage across metagenomes for binning. The authors then analyzed gene content of NCLDV MAGs from both functional and evolutionary standpoints. They concluded that “globally-abundant giant viruses encode complex metabolic capabilities with evolutionary histories largely independent of cellular life”. Given the importance of giant viruses in the ecology of major biomes and evolutionary history of eukaryotes on one side, and scarcity of publically available NCLDV genomes on the other side, I value the relevance of this manuscript. It significantly enhances our understanding of various lineages of giant viruses.

Key points:

The authors adequately provided all data associated with the manuscript, and I could reproduce the high completion estimates of these NCLDV MAGs using a similar methodology based on dedicated HMMs. The text is clear and well written, with the introduction and subsequent sections introducing major publications in the field. The methodology is also adequate. I need to mention here that the metagenomic binning strategy is limited since no differential coverage was used to resolve contigs despite the large amount of metagenomic data available for the surface oceans (biome from which most MAGs were recovered) providing opportunities to do so quite effectively. This is a limitation clearly stated by developers of the binning tool used here (Metabat2), in their publication. This should be acknowledged. However, it is likely that most if not all contig contaminants have been removed by the very careful downstream control quality methodology applied by the authors (ViralRecall and LAST). Thus, it is not surprising that phylogenetic analyses displayed in Figure 5 for few functional genes are so coherent. They support the biological relevance of the findings and suggest a minimal level of contaminants in the database. Finally, the different sections of the results are highly complementary, with a good flow.

Yet, important point and limitations remain to be addressed or acknowledged:

On the front of functional analyses, missing in figure 3B is a column for the reference NCLDVs (regardless of evolution) so that the reader has a perspective of what is new or better emphasized compared to what was discovered before. Please adjust the figure accordingly.

While the concept of virocell is known and accepted by most environmental virologists, it remains rather obscure to non-virologists. Considering the focus of this study and the many mentions of virocell metabolisms, the concept should be introduced more.

This study confirms the large heterogeneity of NCLDV, but suggests a closer relationship between the Mimiviridae and the Phycodnaviridae (Late at the very least). This is reminiscent of a recent publication in PNAS suggesting the existence of higher clades among the NCLDV families (Guglielmini et al., 2019). This study also suggested the ancestry of NCLDV core genes relatively to the emergence of modern eukaryotes, and should be commented concerning the conclusions here on the independent evolution histories of metabolic genes. Similarly, the large detection of Mimiviridae-related MAGs supports conclusions from Mihara and colleagues (2018).

Regarding the identification of clades, authors did not provide information regarding the coherent signal of concatenated genes when used individually and thus it is possible that depending on the phylogenomics strategy (which set of markers) other clades would have emerged. This should be acknowledged.

On a similar front, while the choice of the VT model of substitution has been based on automatic model tests, it remains quite uncommon. Among the empirical stationary models, the more recent LG model would have been more appropriate for comparison with other studies and with the phylogenies of metabolic genes. Ideally, the authors should reconstruct their supergene reconstruction with a mixture model, such as the C60 model, which is included in IQ-TREE but not tested with ModelFinder. While not at the heart of the manuscript, the potential impact of choice of evolutionary model on subsequent identification of clades should be discussed.

Small points:

Ln 42-43: Unclear. Please consider rephrasing. Also, what is a “mode” of evolution?

Ln 71: Authors should clearly state that each MAG was characterized in the context of a single metagenome, without use of differential coverage. Authors might follow up describing the contamination removal step they performed, that likely removed contaminants.

Ln 225: It is only a potential ability. Please rephrase accordingly.

Ln 263: “must” seems a too strong word. Please consider using another one, such as “should”.

Ln 334: please add citation for CheckM and if possible add alternative tool that allows estimation of completion and redundancy for Bacteria and Archaea.

In figure 4 panel D, please highlight the family of MAGs using the same color as in panel A.

Tom O. Delmont

Reviewer #1 (Remarks to the Author):

The authors describe important and novel work in the role of nucleocytoplasmic large viruses (NCLDV) in driving host metabolic processes across a wide range of pathways including nutrient uptake, light-harvesting, carbon metabolism, and nitrogen metabolism. The work is well-grounded in the viral literature and presents a new bioinformatics workflow for binning and distinguishing NCLDV contigs from the cellular metagenomes from which the contigs are derived. Moreover, careful attention is given to quality control, bioinformatics approaches and tool parameter choices for detecting viruses, making the findings compelling and grounded in a solid framework. The work greatly expands the number of available giant virus genomes, and also our understanding of the various clades of giant viruses and their evolutionary context. Finally, the results provide clear evidence for the role of giant viruses as drivers for biogeochemical cycles based on metabolic genes acquired from their hosts in the distant past. The manuscript is clearly written and offers new insight into the global impact of giant viruses that will clearly be well-cited and widely read by a broad audience. I have a few minor points below, and some points to consider in the discussion of the role of giant viruses in the carbon cycle based on my interpretation of the data that the authors may want to consider.

Minor points:

Line 128: A reference is needed after “that has been observed in other studies”

A reference has been added.

Line 411: The authors need to have a reference to where the python script is located in Github to make the code available to the community.

A link with code availability on GitHub is now provided (line 447).

Line 432: The authors need to state which tool was used for comparing representative sequences to reference databases. An e-value cut off is listed by no reference to the tool.

HMMER3 was used- this information is now provided.

Discussion of central carbon metabolism lines 209-226:

The authors show in Figure 3, 4, and S5 that NCLDVs play an important role in carbon metabolism and encode many enzymes in glycolysis, gluconeogenesis, the TCA cycle, and the glyoxylate shunt. However, based on recent literature, the authors may be missing some important and new mechanisms that these viruses may be using that have not been seen in bacteriophage. In particular, the authors should consider these data in conjunction with other findings that “giant virus replication may occur under high oxidative stress” line 164, and “recent studies on model NCLDV-host systems have pointed out ... genes involved in rewiring key aspects of cell physiology during an infection such as apoptosis, nutrient processing, and acquisition, and oxidative stress regulation” (line 159).

First, Figure 4A shows an abundance of succinate dehydrogenase, that has been shown to be involved in the formation and elimination of reactive oxygen species (see <https://doi.org/10.1016/j.bbabc.2016.03.012>). Therefore the role of SDH may not be in central carbon metabolism, but rather in dealing with oxidative stress.

Second, Figure 3b, shows that MM09 has the most carbon metabolic genes (in particular SDH and GAPDH) and also has PhoH related to phosphate acquisition which may be due to limited nutrient availability. Under starvation conditions, GAPDH has recently been shown to play an important role in energy homeostasis (Nature. 2018 Sep;561(7722):263-267. doi: 10.1038/s41586-018-0475-6. Epub 2018 Sep 12). Quoting from that paper, “Mechanistically, this involves the activation of AMPK to induce the redistribution of cytosolic GAPDH to multiple membrane compartments, which then targets ARF GAPs to exert inhibition on different transport pathways (Fig 4q). As the cell encounters starvation in many settings, considerable effort has been devoted to elucidating mechanisms of cell survival during this daunting situation. Autophagy has emerged as a major mechanism 1. In the overall energy equation, autophagy acts on the supply side to increase energy availability. In contrast, the transport inhibition by GAPDH that we have uncovered acts on the demand side to reduce energy consumption. Despite being fundamentally distinct, the two mechanisms nevertheless funnel into a common goal of promoting energy homeostasis during starvation. Another notable distinction between the two mechanisms is that the transport inhibition by GAPDH occurs more rapidly than autophagy. This more rapid mechanism likely allows the cell to stave off the lethal consequences of starvation while longer-term solutions, such as autophagy, are being mobilized.” Therefore, an alternative hypothesis is that NCLDVs produce GAPDH to “stave off the lethal consequences of starvation” in their hosts while autophagy mobilizes amino acids from the host that can be used for viral replication. This is further supported by the abundance of transport-related proteins encoded in these viruses, which may help to promote transport important for viral replication. This could be an important, and currently, an undescribed mechanism that the authors may want to consider especially given the genomic context that they have already laid out in the paper, and the fact that enzymes in C metabolism are not evenly distributed (Figure 4).

We thank the reviewer for this suggestion regarding alternative roles of some of the central carbon metabolism genes in the new giant viruses we report. In the light of new discoveries regarding divergent roles of some carbon metabolism genes in the living cells, we have now added the following discussion in the manuscript (Lines 238-253):

“Although the prevalence of genes involved in central carbon metabolism in the NCLDV MAGs strongly implicates them in modulating host metabolism, it is unclear at this point if these enzymes function in the same physiological context as their corresponding host versions. For example,

succinate dehydrogenase has an important role in modulating cellular oxidative damage ⁴², and could have a similar function during NCLDV propagation, which is carried out within a highly oxidative cellular environment. Moreover, in a recent study GAPDH was directly implicated in starvation-induced negative regulation of vesicle formation in the Golgi and several other cellular transport pathways independent of glycolysis ⁴³. It was proposed that by reducing energy consumption during starvation, GAPDH plays a complementary role in energy homeostasis alongside autophagy - which, in contrast, increases energy availability. Although the modulation of autophagy during NCLDV infection remains to be elucidated, it is possible that viral GAPDH could help avoid the lethal consequences of starvation in the hosts, while autophagy-mediated recycling of proteins makes amino acids and other nutrients available. This possibility is further strengthened by the fact that a large number of NCLDV MAGs harbor phosphate starvation inducible protein (PhoH) and nutrient transporters which might work in concert with the GAPDH-mediated mechanism to ensure virus propagation within the energy-limited host cells”

Reviewer #2 (Remarks to the Author):

In the manuscript entitled “Dynamic Genome Evolution and Complex Virocell Metabolism of Globally-Distributed Giant Viruses”, Moniruzzaman et al. generated 501 metagenome-assembled genomes of Nucleocytoplasmic large DNA viruses (NCLDV) from publically available metagenomes, using automatic binning taking advantage of sequence composition and within-sample coverage. Authors elected not to take advantage of differential coverage across metagenomes for binning. The authors then analyzed gene content of NCLDV MAGs from both functional and evolutionary standpoints. They concluded that “globally-abundant giant viruses encode complex metabolic capabilities with evolutionary histories largely independent of cellular life”. Given the importance of giant viruses in the ecology of major biomes and evolutionary history of eukaryotes on one side, and scarcity of publically available NCLDV genomes on the other side, I value the relevance of this manuscript. It significantly enhances our understanding of various lineages of giant viruses.

Key points:

The authors adequately provided all data associated with the manuscript, and I could reproduce the high completion estimates of these NCLDV MAGs using a similar methodology based on dedicated HMMs. The text is clear and well written, with the introduction and subsequent sections introducing major publications in the field. The methodology is also adequate. I need to mention here that the metagenomic binning strategy is limited since no differential coverage was used to resolve contigs despite the large amount of metagenomic data available for the surface oceans (biome from which most MAGs were recovered) providing opportunities to do so quite effectively. This is a limitation clearly stated by developers of the binning tool used here

(Metabat2), in their publication. This should be acknowledged. However, it is likely that most if not all contig contaminants have been removed by the very careful downstream control quality methodology applied by the authors (ViralRecall and LAST).

Thus, it is not surprising that phylogenetic analyses displayed in Figure 5 for few functional genes are so coherent. They support the biological relevance of the findings and suggest a minimal level of contaminants in the database. Finally, the different sections of the results are highly complementary, with a good flow.

Yet, important point and limitations remain to be addressed or acknowledged:

On the front of functional analyses, missing in figure 3B is a column for the reference NCLDVs (regardless of evolution) so that the reader has a perspective of what is new or better emphasized compared to what was discovered before. Please adjust the figure accordingly.

We have added the numbers of reference genomes and NCLDV MAGs (this study). We feel it would be confusing to add another bubble column, since reference genomes are already represented in some clades and would therefore be counted twice, and we have instead included these values in brackets next to each gene name.

While the concept of virocell is known and accepted by most environmental virologists, it remains rather obscure to non-virologists. Considering the focus of this study and the many mentions of virocell metabolisms, the concept should be introduced more.

We agree with the reviewer on providing more details on the virocell concept and have added the following information in the manuscript (lines 57-60):

“A virocell is defined as a cell undergoing lytic virus infection that has altered nutrient dynamics compared to a healthy cell and a distinct physiological trajectory geared towards virus production rather than cellular growth and propagation¹⁴.”

This study confirms the large heterogeneity of NCLDVs, but suggests a closer relationship between the Mimiviridae and the Phycodnaviridae (Late at the very least). This is reminiscent of a recent publication in PNAS suggesting the existence of higher clades among the NCLDV families (Guglielmini et al., 2019). This study also suggested the ancestry of NCLDV core genes relatively to the emergence of modern eukaryotes, and should be commented concerning the conclusions here on the independent evolution histories of metabolic genes. Similarly, the large detection of Mimiviridae-related MAGs supports conclusions from Mihara and colleagues (2018).

We thank the reviewer for this thoughtful comment. We have added the following discussions in the revised manuscript, with the recommended citations afterwards:

Lines 89-91:

“The identification of large number of *Mimiviridae* members in our study is consistent with previous analyses suggesting high diversity of this family in marine systems ^{10,17,18}.”

And lines 94-97:

“The monophyly of the combined Phycodnaviridae and Mimiviridae families has been suggested previously based on concatenated marker gene phylogenies ^{5,7,19}, although one recent study reported an alternative topology in which the Asfarviridae also placed within this broader group²⁰.”

And lines 279-282:

A recent study suggested that NCDLV have ancient origins and may even pre-date the last eukaryotic common ancestor, lending support to the hypothesis that the initial acquisition of metabolic genes by NCDLV occurred in the distant past ²⁰.

Regarding the identification of clades, authors did not provide information regarding the coherent signal of concatenated genes when used individually and thus it is possible that depending on the phylogenomics strategy (which set of markers) other clades would have emerged. This should be acknowledged.

We have included discussion of this topic in the Methods section under “Phylogenetic Reconstruction of NCDLV MAGs” (lines 451-455):

“Although we used a set of 5 highly conserved markers that have been shown to reliably reconstruct NCDLV evolutionary relationships⁵, other recent studies have used other markers such as DNA-dependant RNA Polymerase (RNAP) subunits and transcription elongation factor II-S (TFIIS)²⁰, and the observed evolutionary relationships between NCDLV groups may be affected by marker gene choice.”

On a similar front, while the choice of the VT model of substitution has been based on automatic model tests, it remains quite uncommon. Among the empirical stationary models, the more recent LG model would have been more appropriate for comparison with other studies and with the phylogenies of metabolic genes. Ideally, the authors should reconstruct their supergene reconstruction with a mixture model, such as the C60 model, which is included in IQ-TREE but not tested with ModelFinder. While not at the heart of the manuscript, the potential impact of choice of evolutionary model on subsequent identification of clades should be discussed.

We reconstructed the phylogeny using the LG+I+G4 model and manually inspected the clades we had defined with the VT+F+I+G4 mode. We observed that 100% of our clades still formed monophyletic groups, supporting our initial clade distinctions. We did observe a small number of topological differences between the LG and VT trees that involved singleton clades (clades with only one member), but these clades still placed reliably in the same broader family (e.g., *Mimiviridae* vs Late *Phycodnaviridae*). The exact phylogenetic placement of these singleton clades does not change the conclusions of our study, and we feel further examination of this subject is outside the scope of the current work. We now provide the LG+I+G4 tree and the original VT+F+I+G4 tree on the FigShare repository for this project, and we discuss these new results in lines 465-468.

“To confirm the clade-level distinctions were consistent across different evolutionary models, we also constructed a phylogenetic tree from the same concatenated protein alignment using the popular LG+I+G4 model and confirmed that our clades still formed monophyletic groups. The LG+I+G4 tree can be found in the supplemental material.”

Small points:

Ln 42-43: Unclear. Please consider rephrasing. Also, what is a “mode” of evolution?

The phrase ‘mode of evolution’ can indeed be confusing, and we have replaced this with ‘the evolutionary processes shaping their genome evolution’. The overall statement has been modified as follows for better clarity:

“The large genomic repertoire of NCLDVs with complex phylogenetic history has sparked interest regarding the evolutionary processes shaping their genome evolution, diversity and potential role in modulating ecological dynamics, which remain poorly understood.”

Ln 71: Authors should clearly state that each MAG was characterized in the context of a single metagenome, without use of differential coverage. Authors might follow up describing the contamination removal step they performed, that likely removed contaminants.

We have added text to clarify these points. Given the complexity of our workflow, however, we feel that most details must be relegated to the Methods section.

Ln 225: It is only a potential ability. Please rephrase accordingly.

We now make it clear these abilities are potential.

Ln 263: “must” seems a too strong word. Please consider using another one, such as “should”.

We have replaced “must” with “should”.

Ln 334: please add citation for CheckM and if possible add alternative tool that allows estimation of completion and redundancy for Bacteria and Archaea.

We now cite CheckM and Anvi'o here

In figure 4 panel D, please highlight the family of MAGs using the same color as in panel A.

This has been done.

REVIEWERS' COMMENTS:

Reviewer #1 (Remarks to the Author):

The authors have addressed all of my comments, and did an excellent job adding to the discussion. This manuscript is ready for publication.

Dr. Bonnie Hurwitz